# Barriers to accessing mental health services for women with perinatal mental illness: systematic review and meta-synthesis of qualitative studies in the UK

Megan Sambrook Smith,[1] Vanessa Lawrence,[2] Euan Sadler,[3] Abigail Easter[3]

¹Global Mental Health, King's College London, Institute of Psychiatry, Psychology & Neuroscience, London, UK
²Health Service & Population Research Department, Institute of Psychiatry, Psychology & Neuroscience, King's College London, London, UK
³Centre for Implementation Science, Health Service & Population Research Department, King's College London, Institute of Psychiatry, Psychology and Neuroscience, London, UK

**Correspondence to**
Dr Abigail Easter;
abigail.easter@kcl.ac.uk

## Strengths and limitations of this study

► This study provides a comprehensive systematic review of barriers to mental healthcare for women with perinatal mental illness, a key public health issue.
► Robust procedures for systematic reviewing and quality assessment were adopted, in line with Preferred Reporting Items for Systematic Reviews and Meta-Analyses reporting guidelines.
► Unidentified barriers, specifically those at structural and organisational levels, may remain due to limited and high-quality research specially looking at perceived barriers at these levels.
► Due to the wide variability in the context of delivery of perinatal mental healthcare globally, this review only included studies conducted with the UK. The findings may, therefore, be less applicable to other healthcare settings.

## ABSTRACT

**Objective** Lack of access to mental health services during the perinatal period is a significant public health concern in the UK. Barriers to accessing services may occur at multiple points in the care pathway. However, no previous reviews have investigated multilevel system barriers or how they might interact to prevent women from accessing services. This review examines women, their family members' and healthcare providers' perspectives of barriers to accessing mental health services for women with perinatal mental illness in the UK.
**Design** A systematic review and meta-synthesis of qualitative studies.
**Data sources** Qualitative studies, published between January 2007 and September 2018, were identified in MEDLINE, PsycINFO, EMBASE and CINAHL electronic databases, handsearching of reference lists and citation tracking of included studies. Papers eligible for inclusion were conducted in the UK, used qualitative methods and were focused on women, family or healthcare providers working with/or at risk of perinatal mental health conditions. Quality assessment was conducted using the Critical Appraisal Skills Programme for qualitative studies.
**Results** Of 9882 papers identified, 35 studies met the inclusion criteria. Reporting of emergent themes was informed by an existing multilevel conceptual model. Barriers to accessing mental health services for women with perinatal mental illness were identified at four levels: Individual (eg, stigma, poor awareness), organisational (eg, resource inadequacies, service fragmentation), sociocultural (eg, language/cultural barriers) and structural (eg, unclear policy) levels.
**Conclusions** Complex, interlinking, multilevel barriers to accessing mental health services for women with perinatal mental illness exist. To improve access to mental healthcare for women with perinatal mental illness multilevel strategies are recommended which address individual, organisational, sociocultural and structural-level barriers at different stages of the care pathway.
**PROSPERO registration number** CRD42017060389.

## INTRODUCTION

Approximately 10%–20% of women experience mental illness during pregnancy or in the first postpartum year (perinatal period).[1–4] Perinatal mental illnesses (PMI) are associated with increased morbidity and are a leading cause of maternal death in high-income countries.[5 6] PMI may also adversely affect psychosocial development of offspring,[7] and are associated with significant long-term socioeconomic costs.[1] Timely identification and treatment of PMI by trained healthcare professionals (HCPs) is paramount.

The 'Five Year Forward View for Mental Health' aims to transform mental health services (MHS) in the UK, and identifies the need to improve perinatal mental health (PNMH) as a strategic priority for the National Health Service (NHS).[8] A key recommendation is 'by 2020/2021, NHS England should support at least 30 000 more women each year to access evidence-based specialist mental healthcare during the perinatal period.'[8] However, in the UK an estimated 60% of

women have no access to PNMH services[2] and 38% of women wait over a month to be referred.[9] Inadequate provision of community MHS, shortages of health visitors (HVs), and midwives and lengthy waiting lists for psychosocial therapies further limit access to MHS for women with PMI.[8]

Barriers to care extend beyond inadequate resources.[10] One survey reported that 30% of women withheld negative feelings from HCPs often due to fear of their baby being taken away.[2] Previous reviews highlight that lack of mother-centred antenatal care, stigmatising attitudes towards mental health and insufficient knowledge among HCPs about PMI contributed to help-seeking delays.[11 12] There is growing evidence suggesting reasons for difficulties accessing MHS are more complex,[13] potentially occur at multiple time points along the care pathway[10] and are compounded by sociocultural and economic issues.[14 15] However, no previous review has synthesised evidence on different stakeholder views of where these perceived barriers exist or how they interact to hinder access to MHS during the perinatal period. Identifying where barriers exist is imperative to developing a comprehensive understanding of how to improve access to PNMH care.

This systematic review and meta-synthesis of qualitative studies in the UK examines perceived barriers to accessing MHS for women with PMI from the perspective of women themselves, their family members and HCPs, and provides evidence to support the implementation of the Five Year Forward View Plan.

## MATERIALS AND METHODS

We conducted a systematic review and meta-synthesis of qualitative studies.

### Search strategy and selection criteria

This systematic review adhered to the Preferred Reporting Items for Systematic Reviews and Meta-Analyses checklist for reporting findings of systematic reviews.

The first author (MSS) initially searched Ovid MEDLINE(R), PsycINFO, EMBASE and CINAHL electronic databases between January 2007 and September 2018 using the following combination of keywords and MeSH terms: ('Perinatal' OR 'Pregnancy' OR 'Birth') AND ('Mental Health' OR 'Mental Disorder') AND ('Health Service Accessibility OR 'Delivery of Health care') AND ('Qualitative Research' OR 'Attitudes of Health Personnel' OR 'Health Knowledge, Attitudes, Practice'), see online supplementary file 1 for the full MEDLINE search strategy used. MSS then hand-searched reference lists of included studies and used citation tracking of these studies in Google Scholar to identify further relevant papers.

We included qualitative studies examining women's, families' and HCPs' perspectives of barriers to accessing MHS for women with mental illness during the perinatal period, published in peer-reviewed English language journals. We defined the perinatal period as any time

from conception to the first-year postnatal. We excluded studies with purely quantitative data or those not conducted in the UK to ensure findings related directly to 'Five Year Forward' implementation. For this review, all mental health conditions which occurred during the perinatal period, including poor general mental well-being and mental 'distress', were included for review. Nicotine addiction and studies exploring barriers to smoking cessation services were excluded from the review.

All papers returned by searches were imported into Endnote (V.X7.7.1) and duplicates removed. The first author conducted initial screening and study selection, then two independent reviewers assessed a random 10% sample (n=19) of full-text search papers for eligibility (agreement measured using Cohen's kappa). Abstracts and titles of each paper were then read and full texts retrieved for studies deemed potentially relevant. Two authors (MSS, AE) discussed studies where inclusion was not clear.

### Quality appraisal

The quality of all included studies was assessed using the Critical Appraisal Skills Programme (CASP) checklist for qualitative studies, which provides a framework for assessing the quality and rigour of selected studies.[16] The CASP provides 10 questions with a series of prompts to guide the assessment of qualitative papers. A scoring system of 1 mark per question was allocated to provide a useful indicator of quality and enabled comparison across reviewers. Each paper was assessed against the CASP tool and a point for each question was allocated if the criteria had been met. The total CASP scores for all papers were then used to categorise studies as either 'strong' (score >9/10; no methodological issues), 'adequate' (score 9–6/10; no major methodological issues) or 'weak' (score <5/10; major methodological issues) quality. Quality assessment was carried out by the first author and then 20% (n=6) by two additional reviewers (ES, AE). Agreement was calculated using Cohen's kappa and disagreements were resolved by discussion between MSS, ES and AE. The use of quality assessments scores is contentious in qualitative research due to difficulties in applying one criterion to multiple qualitative methodologies and journals requesting different reporting requirements. Therefore, we adopted an inclusive approach and studies with a low CASP score were not excluded from the review.

### Data extraction and synthesis approach

A meta-synthesis approach was used to synthesise findings from qualitative studies.[17] The approach was chosen to improve understanding and conceptual development greater than that attained from the individual studies alone.[17] We constructed data extraction tables to record key characteristics and summary findings from included studies (online supplementary table 1). All raw data extracted from each paper were purely in the form of direct quotes and patient sociodemographic data and did

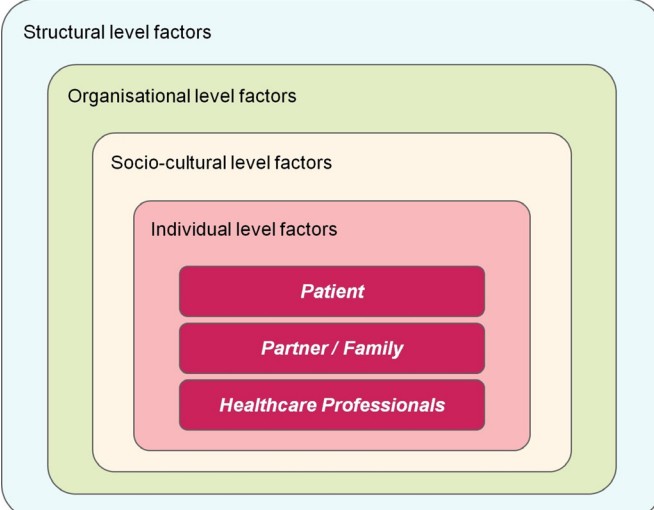

**Figure 1** Adapted model showing multilevel conceptual framework for barriers to mental health services in the perinatal period.[19 20]

not include authors interpretation of their findings. A constant comparison method approach was used to identify emerging themes and related subthemes, including discordant themes, looking for similarities and differences in stakeholder perspectives across the data extracted from all papers. These were then graphically displayed as a 'conceptual map' to visually display themes and explore relationships between themes and related subthemes.[18] All authors met regularly to discuss the emerging themes.

A theoretical multilevel conceptual framework based on the 'delivery systems' model (figure 1)[19 20] was subsequently used to help organise, report and interpret meta-synthesis findings. This adapted model, based on Ferlie and Shortell's 'framework for change'[20] in combination with Reid *et al*'s 'delivery system',[19] was created through discussion between two authors (MSS and ES) after reviewing included papers. This was to allow for specific individual, organisational, sociocultural and structural-level factors (eg, policy and politico-economic factors) to be drawn out of the analysis and provided a theory-driven approach.

### Patient and public involvement
The development of the research questions for this study was directly informed by the NHS 'Five Year Forward View for Mental Health'.[8] The priorities laid out in the Five Year Forward View were established by an independent Mental Health Taskforce, which brought together health and care leaders, service users and experts in the field. The findings will be disseminated widely to service user groups and voluntary organisations.

### RESULTS
In total, we identified 9882 articles, of which 30 qualitative studies met the eligibility criteria. A further five papers were identified through citation tracking. Therefore, a total of 35 papers, reporting on 32 studies, were included for review (figure 2).

Online supplementary table 1 provides a summary table of study characteristics of included qualitative studies. Postnatal depression (PND; n=13) and poor reported mental well-being (n=10) were the most commonly studied PMIs. Other studies commented on rare postnatal outcomes (eg, postpartum psychosis, PP and birth-related Post Traumatic Stress Disorder), antenatal anxiety and perinatal substance misuse. Data collection was mostly via semistructured interviews (n=30) with two studies using non-participant observations and seven studies using focus groups.[10 21–26] In 21 papers, the study population was women with PMI and 13 papers focused on HCPs working with these women, with four studies including family and friends as research participants.

Overall, 88% of the studies were deemed either high (n=18) or moderate quality (n=13). CASP scores below 5 and graded as 'weak' quality (n=4) were associated with poor reflexivity (eg, researchers not considering how their own personal values affected the data collection) and non-rigorous data analysis methods. Online supplementary table 2 provides a full summary of quality ratings for each of the included studies. A low level of agreement in a sample of included studies between reviewers and the original CASP score given by MSS was seen (reviewer 1 (K)=0.2; reviewer 2 (K)=0). This was partly due to poor reporting of information resulting in difficulties assessing the true methodological quality of the studies. Considering these discrepancies, a discussion between MSS, AE and ES took place to reach a consensus on study quality of included studies.

### Meta-synthesis of findings: multilevel barriers to MHS
Barriers to accessing MHS for women with PMI related to a wide range of complex factors. Drawing on Reid *et al*'s delivery system model,[19] such factors operated on multiple levels: individual (knowledge, attitudes and individual characteristics of women, their families and HCPs), organisational (organisational characteristics, service access and inadequacy of resources), sociocultural (family support, wider social support networks and cultural attitudes) and structural (unclear policy) levels.

### Individual-level factors
#### Lack of knowledge about PNMH
Poor PMI awareness and knowledge among HCPs and women was cited in 14 studies as a barrier to accessing appropriate care.[22 27–39] Unfamiliarity with the concept of PNMH and the signs and symptoms of mental illness, as well as a perceived lack of open discussion between HCPs and family members were reported as common issues for women.[22 27 28 30–34 37 38] One woman said:

> I didn't really know the meaning of it [postpartum depression] …I could have detected it earlier if someone had explained to me what your first symptoms were, but nobody told me. (Teenage mother with PND)[28]

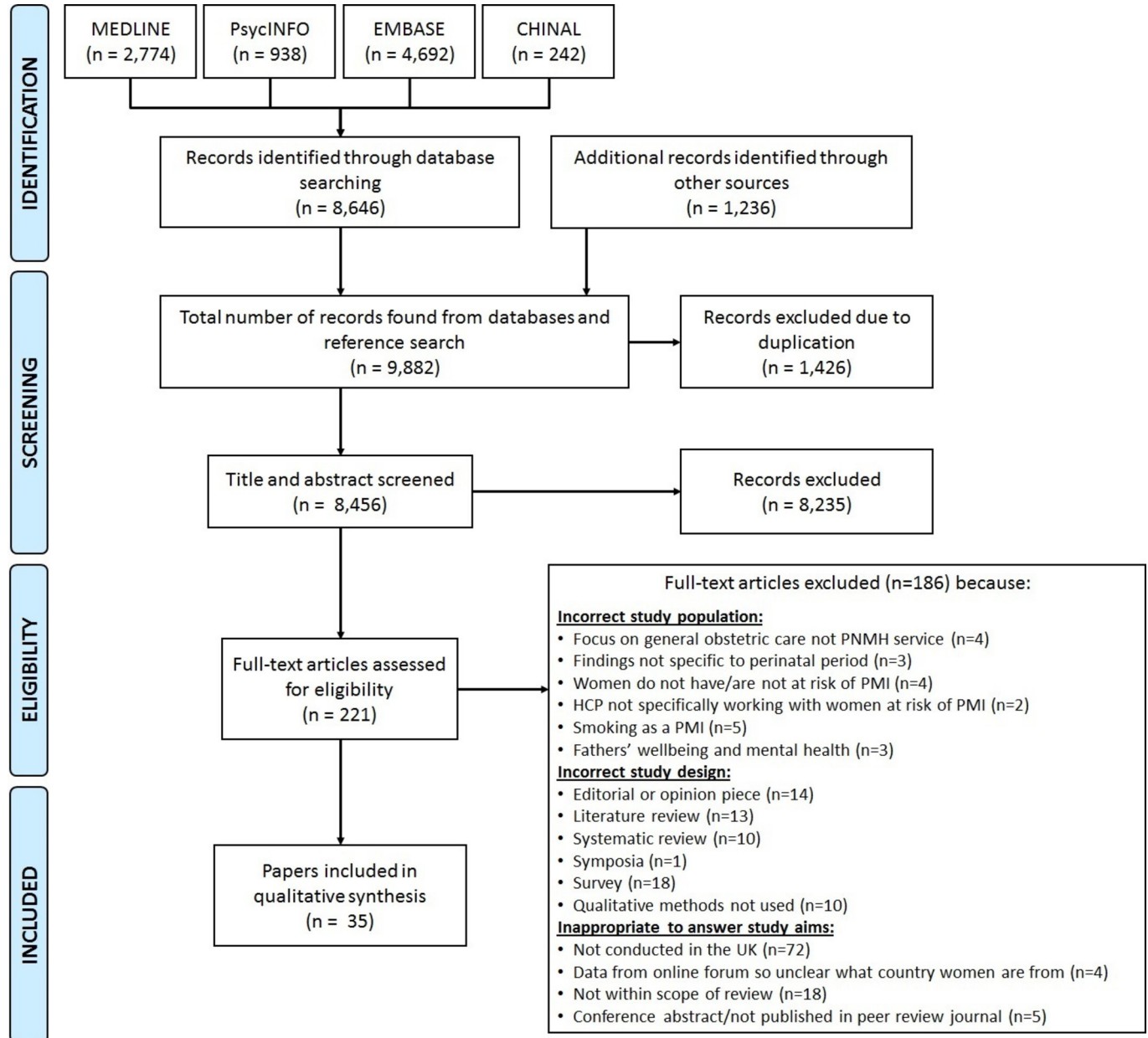

**Figure 2** Study selection. HCP, healthcare professional; PMI, perinatal mental illnesses; PNMH, perinatal mental health.

HCPs similarly reported poor knowledge of PMIs in a number of studies[29 35 36] which was often attributed to inadequate training opportunities.[10 23–25 35 36 40] One student midwife commented that 'mental health is very challenging; we are not trained to give mental healthcare.'[24] Student midwives also highlighted gaps in the training curriculum as 'there was only one lecture on mental health (and) no formal training.'[10 24 39]

Family members and friends of women with PMI played an important role in detecting signs and symptoms of illness. However, several studies found that family and friends could also hinder women from disclosing a mental illness to HCPs, often due to perceived stigma, leading to delays in seeking professional support.[22 27 28 33 34 37 38 41–43] Family members also described feeling unable to recognise deteriorating signs and symptoms and therefore were unable to provide effective support.[29] Normalising symptoms of mental illness due to pregnancy and motherhood was highlighted in several studies as a way of explaining changes in maternal behaviour.[10 27 29 33 43–45] Women with PMI commonly attributed symptoms (eg, low mood and self-esteem) to tiredness or hormones, whereas partners and HCPs tended to dismiss such symptoms as part of the normal pregnancy experience. For example, one male spouse of a woman with PP commented:

> I… didn't really see the more acute signs because A. I'm not experienced in them and B. I knew there was something up but I put it down to her being absolutely over exhausted.[27]

Insufficient knowledge among HCPs about care pathways for women with PMI was reported in 11 studies.[22 26–31 35 46–48] In three of these studies, midwives and HVs voiced a perceived lack of confidence, knowledge and skills to refer women to appropriate services[10 21 23] and obstetricians in one of the studies spoke about not knowing what local services were currently available,[10] resulting in perceived delays in accessing services especially in complex cases[23] and emergencies.[10 25 27]

### Negative attitudes towards mental illness

Stigma, guilt and shame associated with being given a PMI diagnosis and treatment was reported by women in 11 papers.[23 30 32 33 40 42 44 46 49–51] Sometimes women expressed feeling guilty about being ill at a time when happiness was expected.[38 44 50] Commonly concerns centred around negative consequences and stigma of disclosure, such as being labelled a 'bad mum',[30 38 42–45 50] not wanting to upset other family members[34 38] or not fulfilling perceived social expectations of motherhood.[28 30 34 36 37 40 47 52] Women from minority groups particularly felt they were at risk of stigmatising attitudes from HCPs and the public due to cultural differences in social expectations.[22 28 30 37 49] For example, one Pakistani woman with PND said:

'There is a huge stigma of being mentally ill in the public, but for us Asians there is a double disadvantage. I really fear that work will find out.'[37] Similarly, HCPs were sometimes reluctant to formally recognise symptoms related to PND because they did not want to impose labels on women.[10 29] This was emphasised in one study among midwives who reported feeling uncomfortable about recording such concerns in women's medical notes which family members potentially had access to.[24] Furthermore, HCPs in six studies reported that women had refused treatment because of concerns around taking psychotropic medications, including the perceived stigma and feelings of failure as a good mother, fear of harm to their babies and fears of dependence on medications and associated side effects.[22 23 29 30 42 48]

## Organisational-level factors
### Inadequate resources

Inadequate resources in terms of staff shortages and limited service provision were reported by HCPs as key organisational barriers to providing effective services for women with PMI in a number of studies.[21 22 24 36 39 40 42 46] Midwives spoke about not having sufficient time to build rapport with women with PMI and some were 'criticised as slow' by other HCPs if they were perceived to take more time.[24] In one study, a student midwife felt even if 'information and knowledge can be there, there is no time' to provide support for women to access PNMH services, which shaped a sense of frustration.[24] Other logistical barriers related to organisational factors included limited childcare facilities and integration of babies within the therapy session resulting in non-attendance at

appointments.[45 48] There seemed to be mixed responses from health professions as to why the babies could not be integrated within the therapy sessions:

If you're doing some sort of therapy, perhaps trauma work, I don't think it would be appropriate to have a baby in the session because the mum's going to get so upset (CBT therapist)[45]

### Fragmented services: role clarity and conflict

Perceptions of poor continuity of care and not knowing which HVs to contact were reasons for non-disclosure among women with PMI in five studies.[10 28 29 36 40] Some HCPs described how a perceived lack of specialist services and long waiting lists adversely affected access to appropriate care.[10 25] In one further study, variations in service organisation across different NHS Trusts in the UK were viewed by a range of HCPs to cause particular challenges in the healthcare system, and were perceived to compromise the creation of a 'completely secure safety net' of care.[35] For example, one general practitioner (GP) commented:

We have terrible trouble with HVs… because the HVs are now sectorised, we have to liaise with about 12 different HVs. It is just a nightmare! Deeply unsatisfactory! It's not the HVs' fault—it's the system.[21]

Perceptions of fragmented services among HCPs were considered to cause problems with interdisciplinary communication between professional groups, which hindered access to care for women with PMI.[21 23 25 26 29 36 37] Communication was seen as particularly poor between primary care staff and MHS,[25 29 37 42] in emergency situations[10 27 42] and during the handover of care from midwives to HVs.[23 36] This left HVs in one study feeling frustrated and unsupported by other colleagues.[23] HVs in two further studies emphasised how fragmented services created confusion about the HV's role within the referral pathway.[40 42] Similarly, women were also confused about the HV role in supporting them to access appropriate care,[40 42] particularly in terms of liaising with social care providers. Women in a further study voiced uncertainty regarding knowing who was the most appropriate HCP to approach to access PND services.[37] For example, one woman with PND said:

My GP says go the HV and HV says go to GP. I don't know what to do, I need help, don't know where to go, or who to turn to.[37]

## Sociocultural-level factors
### Language barriers

Language as a barrier to accessing MHS and care was similarly reported by both mothers and HCPs in a third of included studies.[21 22 30 31 34 37 41 48 49 53] Women from minority ethnic backgrounds felt they encountered significant barriers when requesting translators.[34 53] For example, one Chinese woman said:

When the midwife visits, I can only speak the sentences about requesting a translator ... They said that this kind of service is limited ... that is what is difficult being Chinese—language barrier. [34]

In one study midwives and HVs seemed to underestimate the importance of translators for such women or were frustrated at the extra work required to arrange such services. [53] This sometimes resulted in the over-reliance on partners of women with PMI for translation which resulted in inaccuracies and ambiguity with exactly what the women wished to communication. [53] For example, one HV commented:

Because sometimes they say loads and then they come back saying, 'She said no'. I know that they've probably done it in shorthand. (HV) [53]

### Differences in cultural values
The relationship between cultural attitudes, access to MHS and associated challenges this raised for women with PMI was an emerging theme in several studies. [22 30 31 34 37 41 48 49 53] The main barriers to accessing appropriate care for women from black minority ethnic (BME) groups included dismissing mental health as a 'something the doctors made up', [30] being unable to disclose feelings due to differences in ethnic backgrounds of HCPs [22 30 31 49] and not receiving perceived culturally appropriate support (eg, no available female doctors). [37] With regard to the need to access specialist services, some women from minority ethnic groups in three studies spoke about the importance of the cultural competency of HCPs to promote and encourage help seeking. [30 36 53] For example, one woman felt that she was met with culturally insensitive attitudes from her consultant:

I went to see the consultant about my hypertension a couple of weeks ago...and when I told him [about HV's 'diagnosis'], he said, 'you haven't got postnatal depression. You're too cheerful and bright and laughing' (BME woman with PND) [49]

### Structural-level factors
#### Unclear policy around appropriate and acceptable use of assessment tools
A key theme among HCPs was the need for clearer policies to be implemented to address potential barriers to accessing MHS's for women with PMI. Polices discussed in various papers centred on the recommended use of appropriate assessment tools for diagnosis of PMI [10 21 23 25 26 29 35 46 51 53] and pathways of care. [10 21 23 45] HCPs frequently expressed negativity towards the use of existing assessment tools (such as the Edinburgh Postnatal Depression Scale). [10 21 23 25 29 35 46 51 53] Midwives, HVs and GPs agreed that such screening tools were currently unsatisfactory, [21 23] and inconsistent usage was perceived to result in many women being missed in the system. [46]

In contrast, women with PMI in four studies found the process of assessment therapeutic as they felt their symptoms were being taken seriously and had received formal recognition from professionals that they were unwell. [40 45 47 53] However, poor implementation of assessment tools by HCPs shaped negative perceptions among women in several studies of the care received. These included feeling that assessments were tick box exercises, [45 46] conducted at inappropriate times, [40 46] and that findings sometimes did not reflect their experiences of PMI. [45 46] Lack of resources, treatment options and poor knowledge of referral pathways also led to perceived ethical concerns among one HV who said:

In an ideal world we'd want to pick them up and then offer them more support, but we can't do that. So there's almost this ethical dilemma of well is there any point in identifying them if you can't do anything with them other than send them to the GP for antidepressants, which isn't good, you know? (HV) [29]

## DISCUSSION
This review has identified multilevel barriers to accessing MHS for women with PMI in the UK during the perinatal period. In summary, we found that negative attitudes towards diagnosis and treatment of PMIs resulted in women avoiding help seeking and reinforced feelings of stigma and guilt. Lack of PNMH knowledge among HCPs, women and their families led to poor recognition of symptoms, delayed referrals and confusion over the role of the HV. Organisational-level factors such as inadequate resources, fragmentation of services and poor interdisciplinary communication compounded these individual-level issues. Structural factors (especially poor policy implementation) and sociocultural factors (eg, language barriers) also caused significant barriers to accessing services for this group of women.

Based on the findings from this review, we propose a conceptual model to explain where these barriers fit within the care pathway for perinatal woman requiring MHS (figure 3).

The first stage of the care pathway involves identification of high-risk women and provision of general PNMH information to all pregnant women. We found that a key barrier to implementing this is poor general knowledge and education about MHS among women with PMI, their families and HCPs, [22 27–39] especially among teenage, [28] BME [21 22 30–32 49] and South Asian mothers. [37 41 48] Evidence has shown that midwives, who are best placed to discuss PNMH risk with pregnant women, receive inadequate PNMH education and training [54 55] with a high proportion reporting receiving no mental health training at all. [54] 'Stepping Forward to 2020', [56] which outlines methods to achieve the 'Five Year Forward View For Mental Health', [8] recommends development and implementation of a competency framework for staff. Our review supports this proposal, however, there is also evidence from our review that women and their families had poor PNMH knowledge, highlighting the need for broader approaches

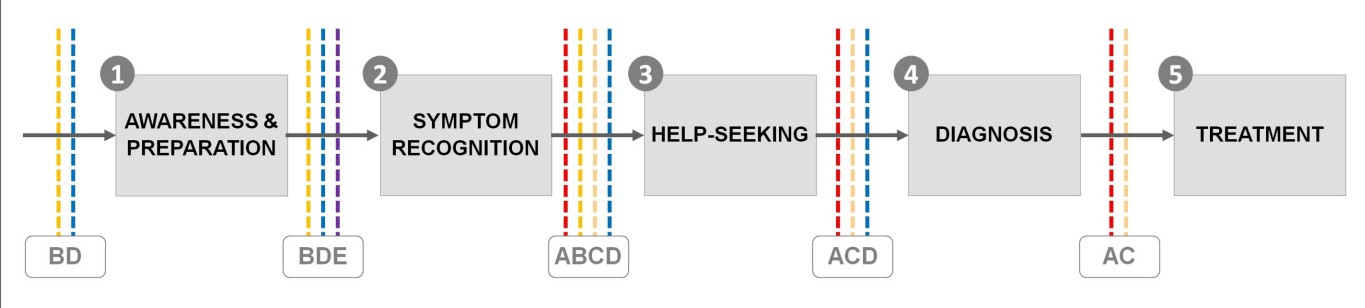

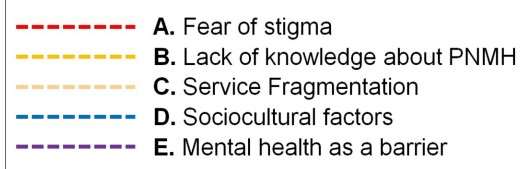

- **A.** Fear of stigma
- **B.** Lack of knowledge about PNMH
- **C.** Service Fragmentation
- **D.** Sociocultural factors
- **E.** Mental health as a barrier

**Figure 3** Conceptual model of key barriers in the care pathway to accessing mental health services during the perinatal period. PNMH, perinatal mental health.

to improve knowledge in these groups. Alongside this, innovative national public health campaigns, such as the Everyone's Business Campaign led by the Maternal Mental Health Alliance, are important for raising awareness and reducing stigmatising attitudes felt by women, which act as barriers at stages 3–5 of the care pathway.

To provide appropriate support HCPs need to correctly identify common PNMH symptoms in distressed women and acknowledge symptoms in mothers who are not actively seeking help via routine mental health assessments (stage 2). Barriers to implementing this identified in this review include difficulties women have in differentiating between PNMH symptoms and 'normal' pregnancy experiences, with some women lacking the insight and capacity to recognise they were unwell, especially those with PP[27 33] and severe PND[29 44]). Family members also play an important role in recognising symptoms of PMI,[22 27 28 33 34 37 38 41–43] however, this was especially difficult for women from minority ethnic backgrounds, for example, those from Chinese,[34] BME[30 31 49] and for South Asian women,[37 41 48] with such subgroups largely not familiar with PNMH and presenting symptoms. High quality and culturally sensitive information about PNMH needs to be provided to each woman to highlight differences between perceived normal pregnancy changes and PNMH symptoms. Information needs to also include red flag signs, information for concerned family members, HCP contact information and emergency protocols. Such resources should be available in multiple languages and adapted for cultural relevance.

Most studies in this review focused on factors influencing help seeking among women with PMI (stage 3). Women often received conflicting advice about who best to approach in the system due to poor interdisciplinary communication between HCPs about their specific roles in the management of women with PMI.[21 23 25 29 36 37] The 'Stepping Forward' report proposes to create new job

roles within PNMH care (eg, psychological well-being practitioner) to address this issue.[56] However, to improve access to services there is also a need for clearer role clarification and understanding of referral pathways.

This review found that barriers preventing diagnosis (stage 4) of PMI among women mostly related to two factors: (1) issues with screening and diagnostic tools; and (2) HCPs reluctance to label mental health conditions due to fear of stigmatisation of the woman with PMI. Both factors, coupled with wider structural and organisational-level barriers (eg, limited resources) shaped complex ethical issues related to the diagnosis and routine screening of PMI for this group of women. The final stage of the care pathway relates to receiving appropriate treatment for women with PMI. Increasing the number of community PNMH services and mother and baby units will provide much needed additional resources;[57] however, it is clear from this review that for women to access these services implementation strategies that address barriers at earlier stages of the care pathway are crucial.

We acknowledge several possible limitations of our review. One possible limitation is that unidentified barriers, specifically those at structural and organisational levels, may remain due to limited research in these areas. Second, although most studies within the review were deemed of 'strong' or 'adequate' quality, there were gaps in reporting especially in terms of under-reporting of possible researcher bias during data collection and analysis. Another potential limitation was that only one reviewer independently reviewed the quality of all included studies (with 10% cross-checked by two reviewers). However, the use of quality appraisal methods in qualitative evidence is contentious and we did not exclude articles on this basis. Furthermore, including papers in the review deemed poor quality did not affect the analysis as extracted themes did not seem to differ

according to CASP scores. Small sample sizes in some of the included studies were another issue in terms of drawing out wider implications. However, the meta-synthesis approach we used enabled the pooling of emerging themes and related subthemes, thus enhancing the robustness and credibility of the results from this review.

In conclusion, this systematic review and meta-synthesis of qualitative studies found multilevel barriers to accessing MHS for women with PMI in the UK. To make tangible and sustainable improvements to expand access to care for this group of patients, we advocate changes need to be implemented at several stages within the proposed care pathway, with specific attention given to targeting key barriers to accessing MHS for women with PMI. Furthermore, in increasing the number of specialist PNMH services and staff, it is also vital that strategies are used to reduce individual, organisational, sociocultural and structural-level barriers that women with PMI are facing in accessing MHS services in the UK. It will not be until these barriers are addressed that the targets outlined in the 'Five Year Forward View for Mental Health' can be optimally met.

**Acknowledgements** The research was supported by the National Institute for Health Research (NIHR) Collaboration for Leadership in Applied Health Research and Care (CLAHRC) South London at King's College Hospital NHS Foundation Trust, and the NIHR Biomedical Research Centre, Guy's and St Thomas' NHS Foundation Trust and King's College London, UK.

**Contributors** AE was responsible for the original conception and design of the work, with significant contributions from MSS, ES and VL. MSS was primarily responsible for conducting the review and data analysis, with quality appraisal checks on a sample of studies conducted by AE and ES. All authors made significant contributions to interpretation of the study findings. MSS produced the initial manuscript draft and further redrafts were critically revised and approved by all authors.

**Funding** AE and ES are funded through individual King's Improvement Science Fellowship awards. King's Improvement Science is part of the NIHR CLAHRC South London and comprises a specialist team of improvement scientists and senior researchers based at King's College London. Its work is funded by King's Health Partners (Guy's and St Thomas' NHS Foundation Trust, King's College Hospital NHS Foundation Trust, King's College London and South London and Maudsley NHS Foundation Trust), Guy's and St Thomas' Charity, the Maudsley Charity and the Health Foundation.

**Disclaimer** The views expressed are those of the authors and not necessarily those of the NHS, the NIHR or the Department of Health.

**Competing interests** None declared.

**Patient consent** Not required.

**Provenance and peer review** Not commissioned; externally peer reviewed.

**Data sharing statement** This study is a systematic review— all data included within the present study have been previously published and in the public domain.

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
