## [Reviewer comments · BMJ Open]

This paper was submitted to a another journal from BMJ but declined for publication following peer review. The authors addressed the reviewers' comments and submitted the revised paper to BMJ Open. The paper was subsequently accepted for publication at BMJ Open.

(This paper received three reviews from its previous journal but only two reviewers agreed to published their review.)

ARTICLE DETAILS

TITLE (PROVISIONAL)	Barriers to Accessing Mental Health Services for Women with Perinatal Mental Illness: Systematic Review and Meta-Synthesis of Qualitative Studies in the UK
AUTHORS	Sambrook Smith, Megan; Lawrence, Vanessa; Sadler, Euan; Easter, Abigail

VERSION 1 – REVIEW

REVIEWER	Jan Taylor Adjunct Associate Professor University of Canberra Australia
REVIEW RETURNED	17-Jul-2018

GENERAL COMMENTS	Thank you for the opportunity to review this excellent paper. It addresses the crucial issue of barriers to accessing mental health services for women in the perinatal period. While the review is situated in research conducted in the UK the findings have application to other countries. I recommend that the authors review the numbers in Figure 2. My calculation of the number of eligible papers is 189 (8054 - 7865). Additionally, I would have appreciated some explanation in the limitations of the decision to include papers with a poor quality CASP rating.
---

REVIEWER	Hanan F. Abdul Rahim Department of Public Health, College of Health Sciences Qatar University Doha, Qatar
REVIEW RETURNED	01-Aug-2018

GENERAL COMMENTS	General Comments This paper is a systematic review and meta-synthesis of qualitative studies on barriers to receiving mental health services in the perinatal period. Two of the distinguishing features of this review are (1) its focus on multiple perspectives of barriers to including care, including mothers, their families, and health care providers and (2) showing how barriers at multiple levels interact to impede women's access to needed mental health services. The originality of this paper and its addition beyond previous reviews (for example see reference 12 cited in this paper) rest on being able to demonstrate the two points clearly. Overall, my impression is that the paper did a better job on the second point (figure 3) than the first in terms of visual (graphical) presentation. I enjoyed reading this review of an important and under-served
---

aspect of perinatal health care. Although the focus of the paper is on the UK, inadequate mental health services in the perinatal period is a problem across many other countries, and the findings will be of interest to readers beyond the UK. The authors make a rigorous effort to summarize qualitative studies and analyze the data using an existing multi-level conceptual framework.

Specific Comments

1. Title: While the title of the paper (Barriers to Accessing Mental Health Services for Women with Perinatal Mental Illness: Systematic Review and Meta-Synthesis of Qualitative Studies in the UK) implies that actual (experienced) barriers will be studied, the objective of the review (stated in the abstract and in the introduction – lines 10-13 and 45-50) states the focus is on perceptions of barriers to receiving mental health services. Though perceptions of barriers can be just as important in terms of affecting access to services, the nuanced distinction should be made.

2. Abstract: eligibility criteria should be mentioned, however briefly, in the structured abstract.

3. Methods:

a. What is the scope of perinatal mental health illness (PMI) referred to by the authors? Can the authors be more specific about the PMIs searched for and covered in this systematic review? Is it non-psychotic mental disorders, for example?

b. Since the authors used Ovid Medline rather than PubMed Medline, can they report more explicitly on the search strategy, including whether they chose to explode MeSH headings and what subheadings were used (if any)?

c. To my knowledge, the CASP Qualitative papers does not suggest a scoring system. Since a score was calculated out of 10, was each question given 1 point? Were all questions weighted as equally important?

d. Only 19 of 188 articles were screened for eligibility by more than one reviewer, and only 6 out of 32 articles were assessed for quality by more than one reviewer. This proportion is quite low in my opinion, especially in light of the low levels of agreement reported (p. 7). Why were more papers not reviewed for inter-rater reliability? I suggest that the reader would want to know more about the items on which most disagreements occurred, to judge the need for further review. The authors are strongly encouraged to include the details of quality assessment in a separate table, which is more informative than (adequate, strong, and weak).

Results

a. Full text articles screened for eligibility should be 189 according to the flowchart and not 188

b. All the findings in this review refer to postnatal depression. While it is known from other reviews that there is a gap in the literature on other perinatal mental health outcomes (for example stress, anxiety, or prenatal depression), I wonder if this is a function of the search strategy used in this paper. In all cases, the authors need to refer to this finding.

c. Generally, there was not enough information on data extraction and synthesis. Which parts of the paper were raw data vs. author's interpretation? To what extent did the authors return to the original context of the paper?

	Minor comments: Table 1:  - Study #1: "provision" not "prevision" in the title of the paper - Study #3: "logistical issues" not "logistically issues" Discussion: The discussion is balanced and thoughtful, identifying some methodological limitations. Ref: Qutteina Y, Nasrallah C, James-Hawkins L, Nur AA, Yount KM, Hennink M, Rahim HF. Social resources and Arab women's perinatal mental health: A systematic review. Women and Birth. 2017 Dec 1.
--	---

REVIEWER	Nicole Highet Centre of Perinatal Excellence
REVIEW RETURNED	08-Aug-2018

GENERAL COMMENTS	This is an important and timely contribution given the focus on perinatal mental health and service access in the UK. Some additional examples of approaches to address some of the barriers (such and innovative approaches to education, destigmatisation campaigns etc.) could be useful in the discussion. I recommend for publishing in current format.
---

VERSION 1 – AUTHOR RESPONSE

Reviewer 1

8. I recommend that the authors review the numbers in Figure 2. My calculation of the number of eligible papers is 189 (8054 - 7865).

Thank you for pointing out this calculation error. The numbers have been corrected and adjusted to account for the new search which identified 3 new papers published between June 2017 and September 2018.

9. I would have appreciated some explanation in the limitations of the decision to include papers with a poor quality CASP rating.

An explanation and justification for adopting an inclusive approach to appraising quality of included papers has been included in the methods section of the paper on page 6. A sentence has also been added into the limitations section to highlight that although papers with lower CASP scores were included, this did not affect the synthesised findings from the review (page 15).

Reviewer 2

10. While the title of the paper (Barriers to Accessing Mental Health Services for Women with Perinatal Mental Illness: Systematic Review and Meta-Synthesis of Qualitative Studies in the UK) implies that actual (experienced) barriers will be studied, the objective of the review (stated in the abstract and in the introduction – lines 10-13 and 45-50) states the focus is on perceptions of barriers to receiving mental health services. Though perceptions of barriers can be just as important in terms of affecting access to services, the nuanced distinction should be made

Thank you for your point highlighting that we should clarify that this review aims to study the perceived barriers, and not necessarily the actual barriers, that women face when accessing

services. This has been clarified in the introduction page 4 to ensure that this is clear to the reader.

11. Eligibility criteria should be mentioned, however briefly, in the structured abstract.

A sentence has been added to the abstract to outline briefly the main eligibility criteria for inclusion

12. What is the scope of perinatal mental health illness (PMI) referred to by the authors? Can the authors be more specific about the PMIs searched for and covered in this systematic review? Is it non-psychotic mental disorders, for example?

This review took a broad approach and did not exclude any papers (except those commenting on nicotine addiction) based on the type of mental health problem experienced by women. All mental health conditions, including psychotic and non-psychotic mental health disorders were included. Two sentences to highlight this have been added on page 5 in the search strategy and selection criteria section of the paper.

13. Since the authors used Ovid Medline rather than PubMed Medline, can they report more explicitly on the search strategy, including whether they chose to explode MeSH headings and what subheadings were used (if any)?

We chose to search the main databases (e.g. Medline, PsychINFO and Embase) using the same software platform (Ovid) because this allowed cross database text word searching (although key words were searched separately using subject headings), and ease of duplication. Although PubMed has some additional content from books and life science journals, we considered that Medline itself had the most relevant content related directly to service provision, mental health and women's experiences, and this content is the same from any provider of the Medline database. This has been clarified through submission of a full search strategy in Ovid Medline as a supplementary file.

14. To my knowledge, the CASP Qualitative papers does not suggest a scoring system. Since a score was calculated out of 10, was each question given 1 point? Were all questions weighted as equally important?

An explanation has been added to the quality appraisal write-up in the methods section on page 6 to clarify the exact methodology of quality appraisal using the CASP tool and how CASP scores were allocated. We have explained that one point was given to each question from this 10question quality appraisal tool in order to directly compare studies despite using varying research methods and reporting structures.

15. Only 19 of 188 articles were screened for eligibility by more than one reviewer, and only 6 out of 32 articles were assessed for quality by more than one reviewer. This proportion is quite low in my opinion, especially in light of the low levels of agreement reported (p. 7). Why were more papers not reviewed for inter-rater reliability? I suggest that the reader would want to know more about the items on which most disagreements occurred, to judge the need for further review. The authors are strongly encouraged to include the details of quality assessment in a separate table, which is more informative than (adequate, strong, and weak).

A table has been added to the supplementary data which outlines the detailed breakdown of CASP scores allocated to each paper and the rationale behind each decision. We have acknowledged that only one reviewer assigning quality assessment scores to all included studies could have been a potential limitation of the review and added this to the limitations section in the discussion on page 15.

16. Full text articles screened for eligibility should be 189 according to the flowchart and not 188

Thank you for pointing out this mistake. The number has been corrected and adjusted to account for the new search which identified new papers published between June 2017 and September 2018.

17. All the findings in this review refer to postnatal depression. While it is known from other reviews that there is a gap in the literature on other perinatal mental health outcomes (for example stress, anxiety, or prenatal depression), I wonder if this is a function of the search strategy used in this paper. In all cases, the authors need to refer to this finding.

The review aimed to make comment on all mental health conditions requiring access to mental health services during the perinatal period and not solely on barriers faced by women with PND. Although most studies in the review includes women with PND, papers also commenting on other conditions (e.g. PP, PTSD and substance misuse) were included for review. Therefore, the themes, findings and discussion of this review do not solely relate to women with PND but paint a broader picture of the system as a whole for women with mental health conditions during the perinatal period and the barriers they are perceived to face at each stage. A sentence to clarify this has been added into the results section, paragraph 2 (page 8). In addition, the full search strategy has been added as a supplementary file to provide further detail on the search strategy used to identify included studies.

18. Generally, there was not enough information on data extraction and synthesis. Which parts of the paper were raw data vs. author's interpretation? To what extent did the authors return to the original context of the paper?

Authors' interpretation of the findings of each individual study was not included within the analysis. Only the raw data, in the form of supporting quotations and patient sociodemographic data, was extracted from each paper for analysis. A sentence has been added to the data extraction and synthesis approach section on page 6 to clarify this approach as part of the metasynthesis methodology.

19. Table 1: Study #1: "provision" not "prevision" in the title of the paper. Study #3: "logistical issues" not "logistically issues"

Thank you for pointing these spelling issues out. These have been amended in the supplementary documents.

Reviewer 3:

10. Some additional examples of approaches to address some of the barriers (such and innovative approaches to education, destigmatisation campaigns etc.) could be useful in the discussion. I recommend for publishing in current format.

Currently our review includes recommendations to broaden educational platforms beyond healthcare staff to patients and their families and the proposal for high quality and cultural appropriate information regarding "red flag symptoms" to be made readily available. We suggest that there is lack of evidence for funding of tertiary services and that there should be funding into the reduction of barriers earlier in the care pathway, and also calls for improvements in role-clarification and established referral pathways. A sentence has been added in the discussion (page 14) to highlight the need for developing innovative anti-stigma campaigns in the UK.

VERSION 2 – REVIEW

REVIEWER	Jan Taylor University of Canberra Australia
REVIEW RETURNED	25-Sep-2018

GENERAL COMMENTS	Thank you for the opportunity to review the revision of this interesting paper. I am satisfied with the authors responses to reviewers comments and the changes they have made.
---

REVIEWER	Hanan Abdul Rahim Department of Public Health, College of Health Sciences, Qatar University
REVIEW RETURNED	01-Oct-2018

GENERAL COMMENTS	I thank the authors for the thoughtful revision of the paper and for the clarity of their responses. I am satisfied that they have addressed the points I raised, and conclude that the paper is acceptable for publication in its current format.
---